# Electroacupuncture Alleviates Depressive-like Behavior by Modulating the Expression of P2X7/NLRP3/IL-1β of Prefrontal Cortex and Liver in Rats Exposed to Chronic Unpredictable Mild Stress

**DOI:** 10.3390/brainsci13030436

**Published:** 2023-03-03

**Authors:** Fang Pang, Yunhao Yang, Siqin Huang, Zhixue Yang, Zhengwei Zhu, Dongmei Liao, Xiao Guo, Min Zhou, Yi Li, Chenglin Tang

**Affiliations:** Chongqing Key Laboratory of Traditional Chinese Medicine for Prevention and Cure of Metabolic Diseases, College of Traditional Chinese Medicine, Chongqing Medical University, Chongqing 400016, China

**Keywords:** depression, electroacupuncture, P2X7R/NLRP3/IL-1β, liver-brain axis, chronic unpredictable mild stress (CUMS), behavior, rat, Baihui (GV20), Yintang (GV29), Ganshu (BL18)

## Abstract

Depression is a complex clinical disorder associated with poor outcomes. Electroacupuncture (EA) has been demonstrated to have an important role in both clinical and pre-clinical depression investigations. Evidence has suggested that the P2X7 receptor (P2X7R), NLRP3, and IL-1β play an important role in depressive disorder. Our study is aimed at exploring the role of EA in alleviating depression-like behaviors in rats. We therefore investigated the effects of EA on the prefrontal cortex and liver of rats subjected to chronic unpredictable mild stress (CUMS) through behavior tests, transmission electron microscopy, Nissl staining, HE staining, immunohistochemistry and Western blotting. Five weeks after exposure to CUMS, Sprague-Dawley (SD) rats showed depression-like behavior. Three weeks after treatment with brilliant blue G (BBG) or EA, depressive symptoms were significantly improved. Liver cells and microglia showed regular morphology and orderly arrangement in the BBG and EA groups compared with the CUMS group. Here we show that EA downregulated P2X7R/NLRP3/IL-1β expression and relieved depression-like behavior. In summary, our findings demonstrated the efficacy of EA in alleviating depression-like behaviors induced by CUMS in rats. This suggests that EA may serve as an adjunctive therapy in clinical practice, and that P2X7R may be a promising target for EA intervention on the liver–brain axis in treatment of depression.

## 1. Introduction

Depressive disorder is a crippling condition that substantially affects psychosocial functioning and reduces life quality [1]. It involves both emotional and physiological components and can cause significant distress and impact daily functioning. Diagnosis requires the presence of symptoms such as persistent low mood or anhedonia. Emotion dysregulation is considered a crucial aspect of depression, as emotions have the ability to influence behavior [2]. Thus, it is essential to comprehend the neural mechanisms behind the impaired inhibitory control, which is prevalent in several psychopathologies and mood disorders, including depression, anxiety, and fear conditioning [3,4]. Emotion regulation has been identified as a central process in both the research and treatment of depression [5].

The intricate interplay between emotions and the brain is facilitated by a number of neural systems spanning from the brainstem to the prefrontal cortex (PFC). The PFC, as a significant nerve center of thinking and behavior regulation in the brain, involves the regulation of emotions, and mediation of cognitive processes, such as the formation of intentions, goal-directed behavior, and attentional control [6], and has emerged as one of regions most consistently impaired in major depressive disorder [7]. The impairment manifests as over or hypo-activation in affective and cognitive tasks requiring emotional or stress regulation, or cognitive control [8]. Recent studies have provided evidence supporting a strong association between mitochondria and the metabolism of kynurenine (KYN). Additionally, it has been observed that malfunction of mitochondria, as well as activation of the tryptophan (Trp)-KYN system, are contributing factors in the development of neuropsychiatric conditions such as depression [9]. Despite extensive research into the neural circuits underlying depression, both in animal models and in human patients, the exact mechanism remains a topic of debate due to the highly heterogeneous nature of depression in terms of its phenomenology, etiology, and pathophysiology [10]. Advances in neuroimaging technology have nonetheless provided valuable insights into the neuroanatomical brain circuits associated with mood disorders [11]. Common themes have emerged, including alterations in volume [12], gray matter density [13] and activity levels across a network of regions including PFC, hippocampus, and amygdala. PFC plays a crucial role in acquisition of fear learning through interactions with the amygdala, hippocampus, and other key neural structures, collectively forming the neural network of fear conditioning [7,14,15,16,17]. Additionally, the relationship between depression and neuroinflammation has been widely recognized [18], as increased microglia activation has been observed in depression-related brain regions, including the PFC [19].

The connection between chronic liver disease and depression has been well established for a long time [20]. As a metabolic disorder, nonalcoholic fatty liver disease and depression share several risk factors, especially chronic systemic inflammation [21,22]. The incidence of depression is three to four times more common in patients with chronic hepatitis than in the general population [23]. Extrahepatic clinical manifestations of chronic hepatitis are commonly associated with the onset of depression—for example, amplified somatic symptoms, exacerbation of functional impairment, and even reductions in treatment adherence and health-related quality of life [24]. Systemic inflammation is pivotal in liver disease, and it is also well documented in patients with depression. Depressive symptoms, behavior, and inflammation are changed by peripheral cytokine signals such as IL-1β via peripheral immune cell-to-brain signaling, notably the activation of macrophages and microglia [25].

In recent years, the activation of microglia and macrophages is mediated by purinergic signaling, which is via the membrane-bound adenosine triphosphate (ATP) receptor, such as the P2X7R [26,27,28]. P2X7R [29] is most commonly associated with activating inflammatory mechanisms in several inflammatory diseases such as liver injury and depression. Furthermore, P2X7R is a critical player in activating the NLRP3 inflammasome, which in turn acts as a signal for the release of IL-1β [30,31]. Interestingly, NLRP3 inflammasome is the pathogenesis of both chronic liver disease and depression [22,32,33]. Based on previous observations [34,35], BBG is a selective and non-competitive P2X7R antagonist with high blood–brain barrier permeability. BBG, meanwhile, is a derivative of a widely used food additive, more than 1 million pounds of which are consumed yearly in the United States [36].

Few researchers have addressed the problem of pathological changes in the liver in patients with depression. The current research focus is not only on the role of EA in the PFC of depressive-like rats but also on pathological changes in the liver. Furthermore, EA may be engaged in the liver–brain axis via P2X7R affecting rats with depressive-like behavior.

In 1996, the World Health Organization added depression to acupuncture indication. Electroacupuncture (EA) combines traditional acupuncture with modern scientific techniques to generate a stable output pattern that overcomes individual differences between therapists. Several existing meta-analyses support EA’s safety and significant clinical efficacy in alleviating depressive symptoms. In addition, patients with depression prefer complementary therapy to drugs when a previous drug treatment has been invalid. Considering the above, our hypotheses are as follows: 1. the physiological function of the liver also alters with depressive-like behavior exposed by CUMS; 2. this alteration is related to the simultaneous activation of P2X7R/NLRP3/IL-1β expression in the prefrontal cortex and the liver; 3. EA alleviates depressive-like behavior and physiological changes in the liver and prefrontal cortex by inhibiting P2X7R/NLRP3/IL-1β expression.

## 2. Material and Methods

### 2.1. Animals and Group Allocation

Rat husbandry and animal strategies were carried out in accordance with the recommendations and protocols authorized by the Institutional Animal Care and Use Committee of Chongqing Medical University. Male SD rats weighing 150–180 g (purchased from the Experimental Animal Center of Chongqing Medical University, Chongqing, China) were used in all experiments. These rats were housed in a temperature-controlled (22 ± 2 °C) and light-controlled (12:12-h light:dark cycle) room and provided with free admission to food and water except on the experimental days. Before the experiment started, the rats were habituated to the experimental conditions for one week. There were control, CUMS, BBG, and EA groups. We used the CUMS [37,38,39] method to establish depressive model rats. First, the rats were randomly separated into two groups: the control group (n = 10) and the model group (n = 80). The two groups of rats were not housed in the same room. During the modeling period, all except the control group were subjected to CUMS for 5 weeks. Because not all rats developed MDD after CUMS, we allocated more rats to the model group. Moreover, the MDD model rats were randomly divided into the CUMS group (n = 10), BBG group (n = 10), and EA group (n = 10). The rats received saline (vehicle for BBG, 0.01% in saline; Sigma) except for the BBG group to which was applied intraperitoneal injections of BBG ((Sigma Aldrich, St. Louis, MO, USA) 50 mg/kg/d) at 9:30 a.m. on 6 consecutive days per week for 3 weeks [40,41] (Figure 1A).

### 2.2. CUMS Procedure

Eighty SD rats in the model group were housed in 80 cages. They were exposed to one of 7 mild stressors daily in a random sequence for 5 weeks [42]. The following stressors were used in the experiment (Table 1).

### 2.3. EA Group

The rats were lightly restrained by hand to minimize stress during EA treatment. Acupuncture needles were inserted bilaterally at the Baihui (GV20), Yintang (GV29), and Ganshu (BL18) acupoints [43,44] to a depth of 5 mm. EA treatment caused slightly visible muscle twitching around the area of insertion. Electrical stimuli were delivered for 20 min at 2 Hz using an Hwato SDZ-III electronic acupuncture treatment instrument. EA treatments were performed from 9:00 a.m.–11:30 a.m. on 6 consecutive days per week for 3 weeks.

### 2.4. Behavioral Tests

The body weight [45] and sucrose preference were tested 9 times from 0th to 8th week; open field test and forced swimming test were carried out only once at the end of the 8th week.

#### 2.4.1. Sucrose Preference Test

Sucrose preference test (SPT) [46] was performed on 4 consecutive days, and included a two-day sugar water adaptation phase, a one-day water deprivation phase, and a one-day test phase. On the first two days (at 8 a.m.), every rat was once simultaneously introduced to two bottles containing 100 mL of either 1% sucrose solution or tap water. The test phase was separated into two parts. At 8 a.m., two bottles were given to each rat, and then after 12 h, the location of the bottles was changed. The rats had free access to food during the test. The test was repeated 8 times with training in between.

#### 2.4.2. Open Field Test

The open field test (OFT) was performed in a 100 cm × 100 cm × 40 cm black plexiglass box with a black floor as previously described [47]. At the beginning of the test, the rats were individually placed in the same location in the corner of the testing box facing the wall. The amount of time they spent in the center zone, the percent of time spent resting, and their average speed of movement in the box were recorded for 5 min, with the data analyzed by means of the SMART video tracking system. After every animal was tested, the equipment used was cleaned with 75% alcohol to eliminate olfactory cues.

#### 2.4.3. Forced Swimming Test

In the forced swimming test (FST) [48,49], the rats were gently placed individually in a 20 cm diameter glass cylinder filled with 23 ± 1 °C water to a depth of 40 cm for 6 min. Data were recorded from the third minute to the end of the trial. Immobility and struggling behavior during the 4 min swimming session were recorded and subsequently analyzed using the SMART video tracking system. Struggling was described as multiple actions of the rat’s forepaws that broke the water, and immobility was used to describe a rat who floated without struggling, solely making those movements essential to preserving the head above the water.

### 2.5. Sample Collection

After the end of the behavior tests, rats from each group were fasted for 24 h. After isoflurane inhalation anesthesia, rats were fixed on the operating table, the chest was opened to expose the heart. A perfusion needle was obliquely inserted into the aorta along the apex of the heart, and the right atrial appendage was open and perfused with 0.09 mol/L PBS solution. When the liquid being pumped from the right atrial appendage became clear, the brain was removed on ice, and the prefrontal cortex and right lobe of the liver were separated. The right brain tissue and right liver lobe of 3 rats from each group were fixed with 4% paraformaldehyde. The remaining prefrontal cortex and liver tissues were quickly placed in a liquid nitrogen tank and then stored in a −80 °C freezer. Three rats from each group were randomly selected, and perfusion with 1% glutaraldehyde solution was performed until the limbs and tail of the rats were stiff. After decapitation, the brains were removed, and the right prefrontal cortex was trimmed to obtain an approximately 1 × 1 × 1 mm sample, and fixed in 2.5% glutaraldehyde at 4 °C.

### 2.6. Transmission Electron Microscopy

The prefrontal cortices were rinsed with 0.1 mol/L phosphoric acid solution 3 times and fixed with 1% osmic acid solution 3 times, then dehydrated in graded alcohol and acetone solutions at 4 °C, and again with 100% acetone at room temperature. After embedding, the tissues were cured in the oven. The tissues were sectioned with an ultrathin slicer at 70 nm thickness and stained with 3% uranium acetate and lead citrate to observe microglia in the prefrontal cortex.

### 2.7. HE Staining

The right lobe of the liver was fixed with 4% paraformaldehyde for 24–48 h, and was then dehydrated in gradient ethanol solutions, embedded in wax, sliced into paraffin sections with a thickness of 5 μm, dewaxed in water, stained with hematoxylin and eosin successively, and sealed. Ten fields were randomly selected from each section, and the liver morphology was observed under a light microscope.

### 2.8. Nissl Staining

Paraffin-embedded tissues were cut at a thickness of approximately 5 μm. After drying, the slides were dewaxed with xylene, dehydrated in gradient ethanol solutions, stained in toluidine blue solution at 56 °C for 20 min, rinsed with distilled water for 5 min, bathed in xylene until transparent for 10 min, sealed with neutral gum, and dried in a ventilated place. Ten fields were observed from each group under an optical microscope, and the results were analyzed by a researcher who did not know the grouping information.

### 2.9. Immunohistochemistry

Paraffin-embedded tissues were cut at a thickness of approximately 4 μm. The sections were dried, dewaxed with xylene, dehydrated in gradient ethanol solutions, and washed with distilled water. Next, the sections were cooled, washed with PBS, incubated with 3% hydrogen peroxide solution for 25 min, washed with PBS, and blocked with 3% BSA at room temperature for 30 min. Paraffin sections of the prefrontal cortex were incubated with an Iba1 (1:1000) primary antibody, and paraffin sections of the liver were incubated with a CD68 (1:500) primary antibody, overnight at 4 °C. The sections were incubated with secondary antibody at room temperature for 50 min after washing with PBS. DAB color development was performed under a microscope after washing in PBS. When the staining was obvious, the sections were washed with tap water, and color development was terminated. After dehydration with anhydrous ethanol and clearing with xylene, the sections were sealed with neutral gum. At least 10 visual fields were randomly observed under a microscope with a 10× lens, and brown CD68- or Iba1-positive cells were observed in the cytoplasm. The optical density of the staining was analyzed with Image Plus software.

### 2.10. Western Blotting

Western blotting was used to measure the expression of P2X7, NLRP3, pro-caspase-1, cleaved-caspase-1, pro-IL-1β, cleaved-IL-1β, and ASC in the prefrontal cortex and liver. Twenty-four hours after the end of the behavioral experiment, the rats were anaesthetized by intraperitoneal injection of pentobarbital sodium., and PBS was perfused through the apex of the heart. The bilateral prefrontal cortex, hippocampus and right lobe of the liver were completely removed. Total protein was extracted from the prefrontal cortex and liver with a tissue protein extraction kit (lysate:phenylmethylsulfonylfluoride = 99:1). The protein concentration was determined by the BCA method, the concentration of each sample was adjusted, and the samples were denatured for 10 min. A total of 25 μg of each protein sample was separated by 12% polyacrylamide gel electrophoresis, and then transferred onto a 0.45 μm or 0.2 μm nitrocellulose membrane. The membrane was incubated with a rabbit anti-NLRP3 monoclonal antibody (3:1000), rabbit anti-P2X7 monoclonal antibody (1:1000), rabbit anti Caspase-1 monoclonal antibody (1:500), rabbit anti-ASC monoclonal antibody (1:1000), or rabbit anti-IL-1β monoclonal antibody (1:1000) overnight at 4 °C, 3 times in TBST for 10 min each the next day, incubated with secondary antibody (goat anti-rabbit antibody, 1:10,000), for 1 h at room temperature, rinsed 3 times with TBST for 10 min each, then developed with chemiluminescence reagent and imaged using an imaging system. Software was used for absorbance analysis.

## 3. Statistical Analyses

All experiments were conducted in a randomized manner. The data sets were analyzed for normality and homogeneity of variance, and parametric post hoc statistical analysis was performed to confirm a priori power calculations. GraphPad Prism 8.0 software (GraphPad, Inc., La Jolla, CA, USA) and SPSS 25.0 (IBM, Armonk, NY, USA) were used for analysis, and *p* < 0.05 was considered significant. The data are reported as the mean ± SD. The data of body weight and SPT data were tested by repeated-measures two-way ANOVA, and the OFT, SPT, immunohistochemistry, and Western blotting data were analyzed by one-way ANOVA. Tukey’s post-hoc test was used to compare the different groups.

## 4. Results

### 4.1. Behavior

The depressive and anxiety-like behaviors—including those in the SPT, FST, and OFT—and body weight of the different groups were examined, to explore the effects of EA on the CUMS rat model. The SPT and body weight measurements were performed nine times, and the FST and OFT were performed only once before execution.

#### 4.1.1. Sucrose Preference and Body Weight

As shown in Figure 1B and Table 2, SPT was not different at baseline. Significant differences in the changes of SPT were found among groups from the 1st week to the 8th week. The repeated-measures analysis of variance showed a statistically significant effect of group (F = 5538.038, *p* < 0.01, *η^2^ =* 0.998), a statistically significant effect of time points (F = 1970.66, *p* < 0.01, *η^2^ =* 0.982), and a statistically significant interaction between time points and group (F = 309.616, *p* < 0.01, *η^2^ =* 0.963).

At the end of the 3rd week of CUMS exposure, the SPT of the CUMS group, BBG group and EA group was decreased by approximately 30%. At the 4th and 5th weeks, the SPT of the CUMS group, BBG group and EA group did not decrease further. At the 6th week, i.e., the week after BBG or EA treatment for 1 week, the SPT of the BBG group was significantly higher than that of the EA group (*p* < 0.05), but there was no difference in SPT between the EA group and the CUMS group (*p* > 0.05). At the 8th week, the SPT of the two groups was markedly higher than that of the CUMS group (*p* < 0.01), and the SPT of the BBG group was clearly greater than that of the EA group (*p* < 0.01). Rats in the CUMS group, BBG group, and EA group displayed decreased sensitivity to reward stimulation and pleasure when they were exposed to CUMS. However, both the BBG and EA exhibited a significant reversal of the decreased sucrose consumption compared to the CUMS group at the 7th week and 8th week of the experiment, as demonstrated by statistically significant results.

The body weight of rats in each group increased, but the body weight gain of the control group was significantly faster than that of the other three groups. Rats exposed to CUMS were observed to have decreased food intake, and showed signs of reduced activity, fur shedding, and reduced luster.

The results of repeated-measures analysis of variance showed a statistically significant effect of group (F = 79.915, *p* < 0.01, *η^2^ =* 0.869), a statistically significant effect of time points (F = 2750.444, *p* < 0.01, *η^2^ =* 0.987), and a statistically significant interaction between time points and group (F = 14.080, *p* < 0.01, *η^2^ =* 0.540). After 5 weeks of CUMS exposure, the weight gain of the rats in the CUMS group, BBG group and EA group was slow, but there was no difference among the groups (*p* > 0.05). After BBG or EA treatment for 1 week, there was no difference in body weight between the rats in the CUMS group, BBG group and EA group, this being significantly lower than the control group (*p* < 0.01). After BBG and EA treatment for 3 weeks, the body weight of the BBG group and EA group was significantly higher than that of the CUMS group (*p <* 0.01), but still significantly lower than that of the control group (*p <* 0.01), and there was no difference in body weight between the two groups (*p* > 0.05), as shown in Figure 1C and Table 3.

#### 4.1.2. OFT and FST

One-way ANOVA showed a significant effect of CUMS on the performance in the FST and OFT (Figure 1, Table 4).

Rats among different groups exhibited significant differences in FST which tests for behavioral despair, including immobility time and total distance. The immobility time and total distance of the CUMS group was significantly greater compared with that of the control group (*p* < 0.01). Compared with the CUMS group, the immobility time and total distance travelled of the BBG group and EA group were significantly lower (*p* < 0.01, *p* < 0.01). Of note, there was no significant difference between the BBG group and EA group (*p* > 0.05), as shown in Figure 1D,E and Table 4.

Results of the OFT showed that compared with the CON group, rats in the CUMS group exhibited significant differences in locomotor activity (percent of time resting, percent of time in center zone, and average speed) (*p* < 0.01). Compared with the CUMS group, the activity ability of the rats in the BBG group and EA group to adapt to a new environment was significantly increased (*p* < 0.01, *p* < 0.01, *p* < 0.01, *p* < 0.01, *p* < 0.01, *p* < 0.01). There was no difference between BBG and EA treatment in the percentage of resting time (*p* > 0.05); however, percent of time spent in the center zone and average speed of the rats in the two groups were notably elevated following the BBG and EA treatment (*p* < 0.05, *p* < 0.01), as shown in Figure 1F–H and Table 4.

### 4.2. Effects of EA on the PFC

#### 4.2.1. Microglial Morphology of PFC

The ultrastructure of the blood–brain barrier and microglial morphology were observed by transmission electron microscopy. In the control group, the blood–brain barrier was intact, microglia were non-oval-shaped, and there was no edema around the cells and blood vessels. In the model group, the blood–brain barrier was loosely arranged, the microglia were oval-shaped, and edema could be seen around the cells and blood vessels. Cell and vascular edema were slightly alleviated in the BBG group and EA group compared with the CUMS group, and microglia were oval shaped in the BBG group and EA group (Figure 2A).

#### 4.2.2. Nissl Staining of PFC

Nissl staining showed that in the control group, neurons exhibited a normal morphology, there were a large number of Nissl bodies, and neurons showed a uniform distribution and dark blue staining. In the CUMS group, the shape of neuronal cells was irregular, the number of Nissl bodies was decreased, neurons were lightly stained, and some Nissl bodies were unevenly distributed. The morphology of neurons in the BBG group and EA group was irregular. Image collection and Nissl body counting were performed by investigators blinded to the group information. The number of Nissl bodies in the CUMS group was significantly lower than that in the control group (*p* < 0.01), and the number of Nissl bodies in the BBG and EA groups was higher than in the control group (*p* < 0.01, *p* < 0.05). However, there was no significant difference in the number of Nissl bodies between the BBG group and EA group (*p* > 0.05). These results are shown in Figure 2B,D and Table 5.

#### 4.2.3. Iba1 Expression in PFC

The results showed that, compared with that in the control group, the expression of Iba1-positive cells in the prefrontal cortex of the CUMS group was increased (*p* < 0.05). Compared with that in the CUMS group, the expression of Iba1-positive cells in the BBG group and the EA group was decreased (*p* < 0.05). These results are shown in Figure 2C,E and Table 5.

#### 4.2.4. Effects of EA on the Expression of P2X7R, NLRP3, and IL-1β Related Protein in PFC

In the prefrontal cortex, the protein expression of P2X7R, pro-caspase-1, cleaved-caspase-1, and ASC were significantly increased in the CUMS group compared with those in the control group (*p* < 0.01), and the protein expression of NLRP3, pro-IL-1β, and cleaved-IL-1β was increased in the CUMS group (*p* < 0.05). Compared with that in the CUMS group, the protein expression of P2X7R in the BBG group was significantly decreased (*p* < 0.01), and the protein expression of NLRP3, pro-caspase-1, cleaved-caspase-1, pro-IL-1β, and cleaved-IL-1β in the BBG group was decreased (*p* < 0.05). The protein expression of NLRP3, P2X7R, pro-caspase-1, pro-IL-1β, and cleaved-IL-1β were decreased in the EA group (*p* < 0.05), but the protein expression of cleaved-caspase-1 and ASC did not change significantly (*p* > 0.05). These results can be seen in Figure 3A–H and Table 6.

### 4.3. Effects of EA on the Liver

#### 4.3.1. HE Staining of Liver

In the control group, liver structure was normal, hepatic lobules were intact, and hepatocytes were of the same size and arranged neatly. There were obvious pathological changes in liver structure in rats in the CUMS group, as hepatocytes were hypertrophic, their arrangement was disorderly, and inflammatory cell infiltration was observed. Liver structure in the BBG group and EA group was basically normal, although there was inflammatory cell infiltration (Figure 4A).

#### 4.3.2. CD68 Expression in Right Liver Lobe

Immunohistochemistry showed that, compared with that in the control group, the expression level of CD68 in the liver tissue of the CUMS group was significantly increased (*p* < 0.01), suggesting that macrophage infiltration was increased. Compared with that in the CUMS group, the expression level of CD68 in liver tissue of the EA group and the BBG group was significantly lower, suggesting that macrophage infiltration was decreased (*p* < 0.05) as shown in Figure 4B,C and Table 5.

#### 4.3.3. Effects of EA on the Expression of P2X7R, NLRP3, and IL-1β Related Protein in Liver

In the liver, the protein expression of NLRP3, P2X7, pro-caspase-1, and ASC were significantly increased in the CUMS group (*p* < 0.01), and the protein expression of cleaved-caspase-1, pro-IL-1β, and cleaved-IL-1β were increased in the CUMS group compared with the control group (*p* < 0.05). Compared with that in the CUMS group, the protein expression of NLRP3, P2X7, pro-caspase-1, leaved-caspase-1, pro-IL-1β, cleaved-IL-1β, and ASC in the BBG group was decreased (*p* < 0.05). The protein expression of NLRP3, P2X7, pro-caspase-1, and cleaved-IL-1β were decreased in the EA group (*p* < 0.05), but there was no difference in the protein expression of cleaved-caspase-1, or ASC (*p* > 0.05). There was no significant difference in NLRP3, P2X7, pro-caspase-1, cleaved-caspase-1, pro-IL-1β, cleaved-IL-1β or ASC protein expression between the BBG group and the EA group (*p* > 0.05) as shown in Figure 3I–P and Table 7.

There was no significant difference in NLRP3, P2X7, pro-caspase-1, cleaved-caspase-1, pro-IL-1β, cleaved-IL-1β or ASC protein expression between the BBG group and the EA group in PFC and liver (*p* > 0.05). The results showed that EA inhibited the expression of inflammatory factors in the prefrontal cortex and liver of rats exposed to CUMS.

## 5. Discussion

Previous research has verified that the procedures used to induce CUMS in rats not only lead to depression-like emotions and behaviors but also accurately simulate the pathological process of depression in a rat model [42,50]. In our study, we aimed to validate this CUMS model and further investigate the potential mechanisms of the antidepressant effect of EA in the PFC and liver. Our findings revealed that CUMS induced depression- and anxiety-like behaviors and caused central and peripheral inflammation. Moreover, our results indicated that EA effectively alleviated these depression- and anxiety-like behaviors, suppressed the expression of P2X7R/NLRP3/IL-1β, reduced the excessive activation of microglia in the PFC and macrophages in the liver, decreased the release of IL-1β, and regulated central and peripheral inflammation. Although few studies have explored the effects of depressive-like models on rat liver cell morphology, our study provides new evidence for the antidepressant effect of EA and offers potential avenues for further exploration in the treatment of depression.

In this study, changes in behavioral assessment at different time points were observed to evaluate the states of nutrition and anhedonia [51,52]. At the start of the study, the four groups at week 0 before intervention showed consistent baseline values. After exposure to CUMS, a significant difference in behavior was observed compared to the control group. The body weight of the CUMS, BBG, and EA groups increased slowly over time, with a significant difference from the CON group by the 8th week. Although a previous study [53] has shown that obesogenic diets can cause depression- and anxiety-like behaviors in rodents, little research has explored the relationship between depression, anxiety and weight loss. Our study indicates that EA effectively regulated the weight loss of CUMS rats and had a positive therapeutic effect. Therefore, we will focus on further indicators to understand the specific mechanisms of the changes in the body weight in future experiments. In previous studies, the SPT was used to evaluate the depression-like behavior in rats through anhedonia, while the FST evaluated the depression-like behavior in rats through despair. Our data showed that exposure to CUMS induced depression-like behavior in the SPT and FST, indicated by decreased sucrose preference and increased immobility time, which was reversed by EA and BBG at the 8th week, suggesting that EA alleviated the CUMS-induced depression-like behavior. Anxiety is a common symptom associated with depression [54]. The OFT is widely used to measure anxiety behavior. Our results indicated that EA alleviated anxiety behavior, such as central zone exploratory behavior.

Previous studies have primarily centered on the role of hippocampal neurons in the development of depression-like behavior in rats and patients with depression [55,56]. The PFC is known to play a critical role in regulating and modifying emotion [57]. Our study and other previous studies have demonstrated that CUMS exposure induces inflammatory injury in the prefrontal cortex [50,58]. The effect of acupuncture to improve PFC function in conditions such as Parkinson’s disease, pain and depression has been well documented [59,60]. Our results are consistent with previous findings [61,62] that indicate EA can effectively inhibit the overactivation of microglia, downregulate P2X7R/NLRP3/IL-1β expression, and reduce IL-1β release in the PFC. Similarly, administration of BBG intraperitoneally produced similar results. This provides basic experimental evidence for exploring the impact of EA on emotion-related neural circuitry.

Chronic liver disease and depression appear to share common risk factors and signaling pathways [63]. Depression is considered a metabolic disease related to liver function, and it has been shown that apolipoprotein B, very-low-density lipoprotein cholesterol, triglycerides, unsaturated fatty acids, tyrosine and abnormal metabolism are related to depression [64,65]. It is often accompanied by liver disease, and can further affect liver function, and even aggravate liver disease [66]. It is widely acknowledged that liver steatosis, characterized by the accumulation of fat in liver cells, often leads to a proinflammatory environment and activates microglial cells [67]. Moreover, it has been suggested that liver steatosis can also trigger a systemic hyperinflammatory state, which can result in damage to the PFC—a phenomenon frequently observed in depression [68]. Our results showed that the morphology of liver cells was significantly altered in CUMS-induced depression in rats and that the increase in the expression of CD68, upregulation of P2X7R/NLRP3/IL-1β expression, and increase in the release of proinflammatory cytokine IL-1β were all improved by EA as well as the intraperitoneal injection of BBG. At present, there are no clinically available drugs that target liver macrophages; thus, these data provide a new idea and strategy for the treatment of depression and liver diseases through targeting liver macrophages. Considering the relationship between liver and PFC function and the occurrence and development of depression, the liver–brain inflammation axis may become a new target for diagnosing and treating depression.

There is growing evidence that stress, both psychological and physical, can activate immune and inflammation processes, which can contribute to the development of depressive symptoms. It is well-established that stress activates microglia, which is a hallmark of neuroinflammation in the central nervous system [27,69]. P2X7R is mainly expressed in microglia [62,70]. P2X7R is the primary driver of inflammation, and the secretion of several proinflammatory cytokines and chemokines depends on the activation of P2X7R by large amounts of ATP released from damaged CNS cells [71]. In particular, P2X7-induced NLRP3 activation has been widely studied in innate myeloid cells (monocytes, macrophages and dendritic cells). Several signaling pathways downstream of P2X7 receptor activation are associated with the induction of NLRP3 inflammasomes [72]. NLRP3 inflammasomes are considered important mediators of depression [73]. NLRP3 inflammasome activation is the pathogenesis of both chronic liver disease and depression. As previously noted, our study and other previous studies have shown that CUMS exposure induces inflammatory injury in both prefrontal cortices [8,50]. In various CUMS-induced models, electroacupuncture has shown a good anti-inflammatory effect [74,75]. Therefore, P2X7R may be a potential target for EA in treating depression.

EA, an integration of traditional Chinese medicine and electronic therapy, has demonstrated efficacy in the treatment of depression and amelioration of depressive symptoms. Based on Chinese medicine theory, “liver controlling dispersion”, EA at Baihui (GV20), Yintang (GV29), and Ganshu (BL18) have positive effects on behavior, the prefrontal cortex, and liver cell function in CUMS-induced depression-like behavior in rats. Baihui (GV20) and Yintang (GV29) are the core acupoints according to the latest research, which is based on data mining technology, on the acupoint characteristics in the treatment of depression by modern acupuncture [76,77]. However, single acupuncture at either GV20 or GV29 fails to alleviate the state of depression [78]. The brain (referred to in traditional Chinese medicine theory as “the spirit’s house”) has the function of regulating memory, feelings, and emotions which is consistent with the theoretical understanding of the role of the cerebral cortex in regulating the human spirit and thinking in modern medicine. The *Shu acu* points, which are located on the back, are the regions where the *qi* of the viscera is infused and are chiefly used to treat disorders of related viscera. Liver dysfunction is the main reason for depression according to TCM. According to clinical evidence [79], we selected the Ganshu (BL18) acupoint. The results of the present study have identified that CUMS-induced depression-like behaviors and that EA at GV20, GV29, and BL18 exhibited antidepressant- and antianxiety-like effects.

Our experiment confirmed our hypothesis, but there are some limitations. The amygdala, hippocampus, thalamus and prefrontal cortex play a synergistic role in cognitive function, learning, memory, emotion, and other functions. However, we did not study in detail the relationship between the hippocampus and prefrontal cortex in depression. As technology continues to advance, the use of brain imaging techniques and real-time brain function recording has become increasingly prevalent in assessing the effects of different therapies on brain function in individuals with depression. This objective evidence provides greater validation for the effectiveness of electroacupuncture (EA) in the treatment of depression. Further investigation is required to determine the extent to which EA can improve depression-related neural circuits through the application of advanced neuroimaging methods, and to evaluate whether EA can enhance liver metabolism and mitigate metabolic changes in CUMS-induced depression-like behavior in rats. While the current findings are derived in a laboratory setting, additional randomized controlled trials will be necessary to establish clinical efficacy.

## 6. Conclusions

In conclusion, the results of the study provide evidence to support the hypothesis that rats displaying CUMS-induced depression-like behavior exhibit damage to the prefrontal cortex and liver, characterized by an inflammatory state triggered by microglia and macrophages. The antidepressant effect of EA may be achieved through its ability to modulate inflammation by downregulating the expression of P2X7R/NLRP3/IL-1β and reducing the release of IL-1β. This study highlights the potential role of EA in alleviating depression symptoms associated with CUMS and provides new experimental evidence for the use of EA as an add-on therapy in the treatment of depression and the co-occurrence of chronic liver disease and depression.

## Figures and Tables

**Figure 1 brainsci-13-00436-f001:**
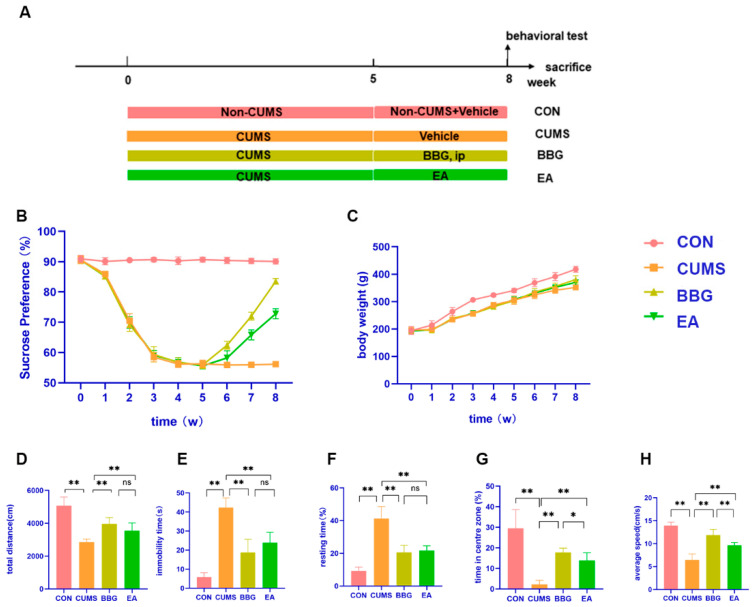
CUMS procedure and effect of EA on body weight gain reduction and depressive, anxiety-like behavior in CUMS rats: (**A**) experimental design, (**B**) sucrose preference test (SPT), (**C**) body weight, (**D**,**E**) forced swimming test (FST), (**F**–**H**) open field test (OFT). * *p* < 0.05, ** *p* < 0.01.

**Figure 2 brainsci-13-00436-f002:**
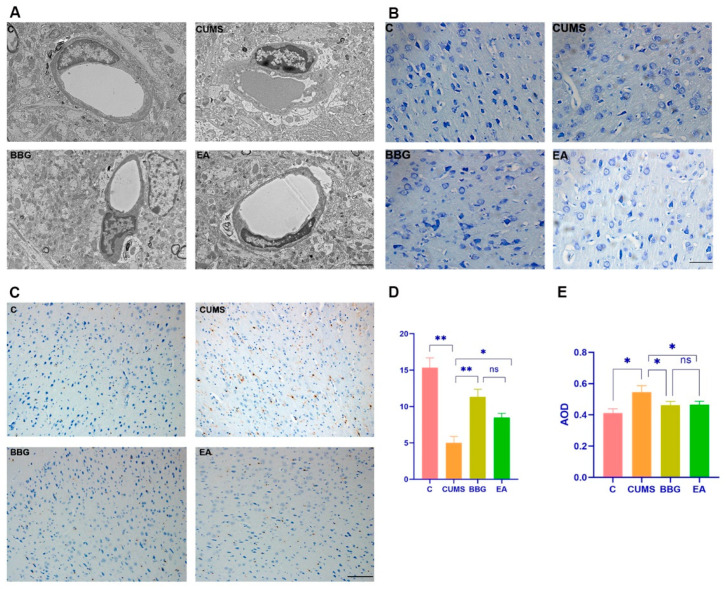
Effects of EA on the PFC: (**A**) microglial morphology, (**B**) Nissl bodies, and (**C**) comparison of Iba1 expression in PFC. Nissl body count of PFC of rats in each group (**D**), comparison of Iba1 expression in PFC in each group (**E**). * *p* < 0.05, ** *p* < 0.01. The scale bars in A = 2 μm, the scale bar in B = 50 μm, the scale bar in C = 100 μm.

**Figure 3 brainsci-13-00436-f003:**
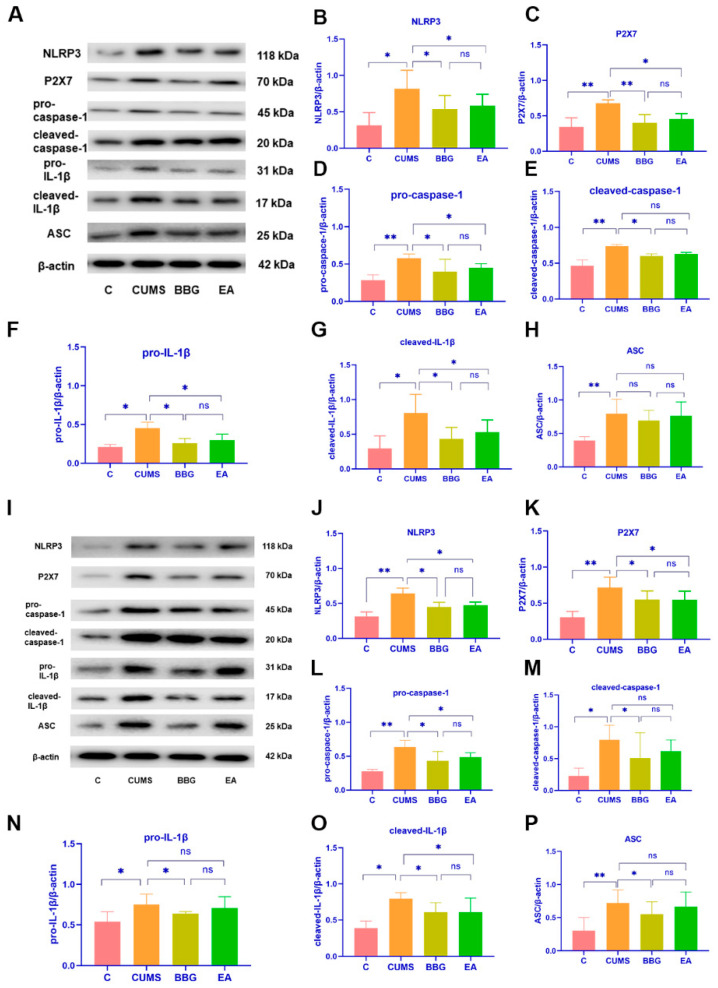
Effects of EA on the expression of the protein of the prefrontal cortex (**A**–**H**), and the liver (**I**–**P**) of rats of each group. Values are presented as the means ± standard error of the mean. N = 6, * *p <* 0.05, ** *p <* 0.01.

**Figure 4 brainsci-13-00436-f004:**
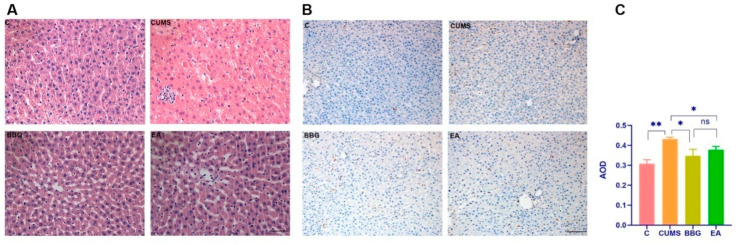
Effects of EA on the liver: (**A**) HE staining of liver, (**B**) CD68 expression in right liver lobe, and (**C**) comparison of CD68 expression in right liver lobe. * *p* < 0.05, ** *p* < 0.01. The scale bars in A = 50 μm, and in B = 100 μm.

**Table 1 brainsci-13-00436-t001:** Daily schedule of the CUMS paradigm.

Content	Day
cold swimming	D1, D4, D10, D19, D25, D34
water deprivation	D2, D15, D24, D28, D32
food deprivation	D9, D11, D21, D29
continuous illumination	D3, D13, D17, D20, D27, D35
tail clamping	D5, D8, D14, D22, D33
tail suspension	D7, D16, D21, D26, D30
wet bedding	D6, D12, D18, D23, D31

**Table 2 brainsci-13-00436-t002:** Effects of EA on the SPT (n = 10; $\bar{x}$ ± s; %).

Groups	0 Week	1 Week	2 Weeks	3 Weeks	4 Weeks	5 Weeks	6 Weeks	7 Weeks	8 Weeks
CON	90.93 ± 1.12	90.13 ± 1.16	90.48 ± 0.96	90.66 ± 0.69	90.29 ± 1.26	90.68 ± 0.80	90.41 ± 1.03	90.24 ± 0.90	90.08 ± 0.87
CUMS	90.56 ± 1.14	85.93 ± 0.66 ****	70.52 ± 2.26	58.48 ± 1.53	56.22 ± 1.18	56.30 ± 1.19 ****	55.91 ± 0.88 ****	55.94 ± 0.80 ****	56.17 ± 0.89 ****
BBG	90.60 ± 1.27	85.16 ± 0.93 ****	69.10 ± 2.16	59.38 ± 2.52	56.69 ± 1.60	55.89 ± 1.17 ****	62.42 ± 1.27 ^◊◊^	72.00 ± 1.32 ^◊◊^	83.49 ± 0.93 ^◊◊^
EA	90.55 ± 1.15	85.11 ± 0.81 **	69.92 ± 1.91	59.14 ± 1.48	56.85 ± 1.60	55.63 ± 1.01 ****	58.22 ± 2.33	62.80 ± 1.65 ^◊◊□□^	72.80 ± 1.63 ^◊◊□□^

Data were expressed as mean ± SD. *** p* < 0.01, compared with the CON group; ^◊◊^
*p* < 0.01, compared with the CUMS group; ^□□^
*p* < 0.01, compared with the BBG group.

**Table 3 brainsci-13-00436-t003:** Effects of EA on the body weight (n = 10; $\bar{x}$ ± s; g).

Groups	0 Week	1 Week	2 Weeks	3 Weeks	4 Weeks	5 Weeks	6 Weeks	7 Weeks	8 Weeks
CON	194.21 ± 14.20	214.02 ± 15.63	263.72 ± 14.22	306.32 ± 6.98	323.77 ± 5.51	340.52 ± 7.85	368.68 ± 14.73	391.24 ± 14.58	418.02 ± 10.78
CUMS	195.68 ± 12.85	198.6 ± 8.31	234.44 ± 5.27	256.03 ± 9.28	287.72 ± 9.30	305.55 ± 15.46 ****	324.46 ± 13.70	341.28 ± 10.76	351.44 ± 8.89
BBG	197.13 ± 4.76	195.18 ± 4.80	238.90 ± 4.25	255.44 ± 4.22	280.82 ± 7.43	303.52 ± 7.03 ****	334.86 ± 11.98	355.22 ± 11.76 ^◊^	380.27 ± 13.64 ^◊◊^
EA	192.94 ± 8.47	196,88 ± 5.90	234.20 ± 2.60	258.20 ± 4.18	281.41 ± 19.08	307.69 ± 5.57 ****	330.50 ± 5.18	352.08 ± 5.79	370.44 ± 8.47 ^◊◊^

Data were expressed as mean ± SD. *** p* < 0.01, compared with the CON group; ^◊^
*p* < 0.05, ^◊◊^
*p* < 0.01, compared with the CUMS group.

**Table 4 brainsci-13-00436-t004:** Effects of EA on OFT and FST (n = 10; $\bar{x}$ ± s; cm, s, %, cm/s).

Groups	Total Distance (cm)	Immobility Time (s)	Resting Time Percent (%)	Time in Centre Zone (%)	Average Speed (cm/s)
CON	5072.25 ± 527.29	6.80 ± 1.36	9.33 ± 2.30	29.56 ± 9.10	13.92 ± 0.77
CUMS	2849.84 ± 193.97 ****	42.33 ± 5.08 ****	41.35 ± 7.26 ****	2.26 ± 1.93 ****	6.45 ± 1.31 ****
BBG	3967.77 ± 379.50 ****^◊◊^	18.78 ± 5.89 ****^◊◊^	20.63 ± 4.26 ****^◊◊^	17.77 ± 2.06 ****^◊◊^	11.87 ± 1.22 ****^◊◊^
EA	35342.04 ± 477.89 ****^◊◊^	23.89 ± 5.51 ****^◊◊^	21.73 ± 2.84 ****^◊◊^	13.84 ± 3.74 ****^◊◊□^	9.63 ± 0.61 ****^◊◊□□^

Data were expressed as mean ± SD. *** p* < 0.01, compared with the CON group; ^◊◊^
*p* < 0.01, compared with the CUMS group; ^□^
*p* < 0.05, ^□□^
*p* < 0.01, compared with the BBG group.

**Table 5 brainsci-13-00436-t005:** Effect of EA on the Nissl body count and expression of Iba1 in PFC and CD68 in right liver lobe of CUMS rats.

Groups	Nissl Body Counts	Iba1 Relative Intensity	CD68 Relative Intensity
CON	15.33 ± 1.36	0.41 ± 0.03	0.31 ± 0.02
CUMS	5.00 ± 0.89 ****	0.55 ± 0.04 ***	0.43 ± 0.01 ****
BBG	11.33 ± 1.03 ^◊◊^	0.46 ± 0.02 ^◊^	0.34 ± 0.03 ^◊^
EA	8.5 ± 0.54 ^◊^	0.46 ± 0.02 ^◊^	0.38 ± 0.02 ^◊^

(n = 10; $\bar{x}$ ± S.E.). Data were expressed as mean ± SEM. ** p* < 0.05, *** p* < 0.01, compared with the CON group; ^◊^
*p* < 0.05, ^◊◊^
*p* < 0.01, compared with the CUMS group.

**Table 6 brainsci-13-00436-t006:** Effects of EA on the expression of P2X7R, NLRP3, and IL-1β related protein in the PFC of CUMS rats.

Groups	NLRP3/β-Actin	P2X7R/β-Actin	Pro-Caspase-1/β-Actin	Cleaved-Caspase-1/β-Actin	Pro-IL-1β/β-Actin	Cleaved-IL-1β/β-Actin	ASC/ β-Actin
CON	0.31 ± 0.18	0.34 ± 0.11	0.28 ± 0.05	0.47 ± 0.06	0.21 ± 0.02	0.29 ± 0.16	0.39 ± 0.04
CUMS	0.81 ± 0.25 ***	0.68 ± 0.04 ****	0.54 ± 0.04 ****	0.74 ± 0.06 ****	0.52 ± 0.05 ***	0.89 ± 0.06 ****	0.80 ± 0.15 ****
BBG	0.53 ± 0.18 ^◊^	0.40 ± 0.10 ^◊◊^	0.40 ± 0.12 ^◊^	0.59 ± 0.05 ^◊^	0.26 ± 0.04 ^◊^	0.41 ± 0.10 ^◊^	0.69 ± 0.11
EA	0.58 ± 0.16 ^◊^	0.45 ± 0.07 ^◊^	0.45 ± 0.04 ^◊^	0.60 ± 0.06	0.28 ± 0.01 ^◊^	0.47 ± 0.05 ^◊^	0.76 ± 0.15

(n = 6; $\bar{x}$ ± s.) Data were expressed as mean ± SD. ** p* < 0.05, *** p* < 0.01, compared with the CON group; ^◊^
*p* < 0.05, ^◊◊^
*p* < 0.01, compared with the CUMS group.

**Table 7 brainsci-13-00436-t007:** Effects of EA on the expression of P2X7R, NLRP3, and IL-1β related protein in the liver of CUMS rats.

Groups	NLRP3/β-Actin	P2X7R/β-Actin	Pro-Caspase-1/β-Actin	Cleaved-Caspase-1/β-Actin	Pro-IL-1β/β-Actin	Cleaved-IL-1β/Β-Actin	ASC/ β-Actin
CON	0.31 ± 0.05	0.30 ± 0.07	0.28 ± 0.02	0.23 ± 0.08	0.54 ± 0.09	0.38 ± 0.05	0.30 ± 0.14
CUMS	0.64 ± 0.07 ****	0.72 ± 0.12 ****	0.63 ± 0.07 ****	0.79 ± 0.16 ***	0.75 ± 0.09 ***	0.73 ± 0.14 ***	0.72 ± 0.14 ****
BBG	0.44 ± 0.06 ^◊^	0.55 ± 0.10 ^◊^	0.43 ± 0.10 ^◊^	0.51 ± 0.28 ^◊^	0.64 ± 0.02 ^◊^	0.58 ± 0.10 ^◊^	0.55 ± 0.13 ^◊^
EA	0.47 ± 0.04 ^◊^	0.55 ± 0.04 ^◊^	0.48 ± 0.05 ^◊^	0.61 ± 0.13	0.70 ± 0.10	0.57 ± 0.07 ^◊^	0.66 ± 0.16

(n = 6; $\bar{x}$ ± s.) Data were expressed as mean ± SD. ** p <* 0.05, *** p* < 0.01, compared with the CON group; ^◊^
*p* < 0.05.

## Data Availability

The data sets used and/or analyzed during the current study are available from the corresponding author on reasonable request.

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
