# Peer review of "Electroacupuncture Alleviates Depressive-like Behavior by Modulating the Expression of P2X7/NLRP3/IL-1β of Prefrontal Cortex and Liver in Rats Exposed to Chronic Unpredictable Mild Stress"

_brainsci, 2023, doi:10.3390/brainsci13030436_

Round 1

Reviewer 1 Report

13 January 2023 

Manuscript ID: brainsci-2175781

Type: Article

Title: ‘Effects of electroacupuncture on prefrontal cortex and liver of rats exposed to CUMS through P2X7R/NLRP3/IL-1β’ by Pang et al., submitted to Brain Sciences 

Dear Authors,

The present research article by Pang and colleagues, entitled ‘Effects of electroacupuncture on prefrontal cortex and liver of rats exposed to CUMS through P2X7R/NLRP3/IL-1β’ is a well-written and useful summary on the status of knowledge on the role of electroacupuncture (EA) in depressive-like mouse model, focusing on the effects of EA on the prefrontal cortex and liver of rats subjected to chronic unpredictable mild stress (CUMS).

The main strength of this manuscript is that it addresses an interesting and timely question, describing how EA might alleviate depressive like behavior and changes in the liver and prefrontal cortex of rats exposed to chronic unpredictable mild stress, by inhibiting P2X7R/NLRP3/IL-1β expression.

In general, I think the idea of this article is really interesting and the authors’ fascinating observations on this timely topic may be of interest to the readers of Brain Sciences. However, some comments, as well as some crucial evidence that should be included to support the author’s argumentation, needed to be addressed to improve the quality of the manuscript, its adequacy, and its readability prior to the publication in the present form, in particular reshaping parts of the Introduction and Methods sections by adding more evidence and theoretical constructs.

Please consider the following comments:

1.      Title: Please present the title self-explanatory and stating the most important message of this study and avoid using abbreviation if possible.

2.      Abstract: According to the Journal’s guidelines, the abstract should be a total of about 200 words maximum. Also, according to the Journal’s guidelines, this section should be presented as a single paragraph, without explicit sub-headings. Please present the abstract, proportionally describing the background, the objectives, the methods, the results, and the conclusion. The background should include the general background (one to two sentences), the specific background (two to three sentences), and current issue addressed to this study (one sentence). The result should include one sentence describing the main result using such words like “Here we show”. The conclusion should write the potential and the advance this study has provided in the field and finally a broader perspective (two to three sentences) readily comprehensible to a scientist in any discipline.

4. A graphical abstract that will visually summarize the main findings of the manuscript is highly recommended.

3.      Keywords: Please consider adding ‘chronic unpredictable mild stress (CUMS)’ as a keyword and list ten keywords and use as many as possible in the title and in the first two sentences of the abstract

4.      In general, I recommend authors to use more references to back their claims, especially in the Introduction of this meta-analysis, which I believe is lacking. Thus, I recommend the authors to attempt to expand the topic of their article, as the bibliography is too concise. Nevertheless, I believe that less than 50 articles are too low for a research article. Therefore, I suggest the authors to focus their efforts on researching relevant literature: in my opinion, adding more citations will help to provide better and more accurate background to this study.

5.      Introduction: The introduction would benefit from a reorganization of the sub-paragraphs. As it stands, there is confusion in terms of the flow of information. I suggest beginning with a theoretical explanation of mood disorders and the role of prefrontal cortex in the pathophysiology of depressive disorder. In this regard, I would suggest adding more information on pathological neural substrates of depression disorder, for example focusing on ‘Dissecting Neurological and Neuropsychiatric Diseases: Neurodegeneration and Neuroprotection’ and on structural as well as functional abnormalities of prefrontal cortex that may affect patients’ cognitive impairments (https://doi.org/10.17219/acem/146756; doi: 10.3389/fnbeh.2022.998714). In my opinion, authors could further explore relationship between the molecular regulation of higher-order neural circuits and neuropathological alterations in this neuropsychiatric disorder (https://doi.org/10.3390/cells11162607) and how they may impact on the ‘Functional interplay between central and autonomic nervous systems in human fear conditioning’.

6.      Materials and Methods: I recommend that the authors present more references to ensure reliability and integrity of evidence in the study design and the methodology the authors have applied to this study.

7.      Results: I suggest rewriting this section more accurately. To properly present experimental findings, I think that authors should provide full statistical details (like degree of freedom or post-hoc utilized), to ensure in-depth understanding and replicability of the findings. Specifically, provide more detail about levels of IL-1β in the prefrontal cortex, because it appears unclear and hard to grasp how to interpret sets of proteins. Also, in my opinion, it is necessary for the authors to present their findings using summary tables.

8.      Discussion: The authors need to totally reorganize and fully expand this section. Starting with the summary of the previous section (Results), the authors need to develop discussion on the potential of this study complementing as the extension of the previous work, the implication of the findings of this study, how this study could facilitate future research, the ultimate goal, the challenge, the knowledge and the technology necessary to achieve this goal, the statement about this field in general, and finally the importance of this line of research. 

9.      In my opinion, I think the ‘Conclusions’ paragraph would benefit from some thoughtful as well as in-depth considerations by the authors, because as it stands, it is very descriptive but not enough theoretical as a discussion should be. The authors should make their effort to explain the theoretical implication as well as the translational application of their research.

10.  In according to the previous comment, I would ask the authors to include a proper and defined ‘Limitations and future directions’ section before the end of the manuscript, in which authors can describe in detail and report all the technical issues brought to the surface.

11.  Tables and Figures: According to the Journal’s guidelines, please provide a short explanatory caption for the table within the text. Also, I suggest modifying all figures for clarity because, as it stands, the readers may have difficulty comprehending it and to change the scale of the vertical axis and use the same minimum/maximum scale value in all the graphs. Also, please present all figures in color.

12.  References: Authors should consider revising the bibliography, as there are several incorrect citations. Indeed, according to the Journal’s guidelines, they should provide the abbreviated journal name in italics, the year of publication in bold, the volume number in italics for all the references.

Overall, the manuscript contains 1 table, 4 figures and 30 references. I believe that the manuscript might carry important value in describing how EA might alleviate depressive like behavior and changes in the liver and prefrontal cortex of rats exposed to chronic unpredictable mild stress, by inhibiting P2X7R/NLRP3/IL-1β expression.

I hope that, after these careful revisions, this paper can meet the Journal’s high standards for publication. I am available for a new round of revision of this paper.

Best regards,

Reviewer

Reviewer 2 Report

In abstract: the first sentence after Method should be in a part of Background.

Author Response

Response: Thank you for your decision and constructive comments on our manuscript. We feel great thanks for your valuable feedback that we have used to improve the quality of our manuscript.

Abstract

Depression is a complex clinical disorder associated with poor outcomes. Electroacupuncture (EA) has been demonstrated to have an important role in both clinical and pre-clinical depression investigations. Evidence has suggested that the P2X7 receptor(P2X7R), NLRP3, and IL-1β play an important role in depressive disorder. Our study is aimed at exploring the role of EA in depressive-like rats. Here we undertook the effects of EA on the prefrontal cortex and liver of rats subjected to chronic unpredictable mild stress (CUMS) through behavior tests, transmission electron microscopy, Nissl staining, HE staining, immunohistochemistry and western blotting. Five weeks after exposure to CUMS, Sprague-Dawley (SD) rats showed depressive-like behavior. Three weeks after treatment with brilliant blue G (BBG) or EA, depressive symptoms were significantly improved. Liver cells and microglia showed regular morphology and orderly arrangement in the BBG and EA groups compared with the CUMS group. Here we show EA downregulated P2X7R/NLRP3/IL-1β expression and relieved depressive-like behavior. In summary, our findings demonstrated the efficacy of EA in alleviating depressive-like behaviors induced by CUMS in rats. This suggests that EA may serve as an adjunctive therapy in clinical practice, and P2X7R may be a promising target for EA intervention on the liver-brain-axis in depression treatment.

Reviewer 3 Report

The Manuscript: „ Effects of electroacupuncture on prefrontal cortex and liver of rats exposed to CUMS through P2X7R/NLRP3/IL-1β’’ by Fang Pang and colleagues explored the role of EA in depressive-like rats. Tthe effects of EA on the prefrontal cortex and liver of rats subjected to chronic unpredictable mild stress (CUMS) were analysed through behavior tests, transmission electron microscopy, Nissl staining, HE staining, immunohistochemistry and western blotting. After going through the manuscript, I have a couple of comments for the authors:

1.     Did EA treatment also alleviate the microglial activation in the hippocampus?

2.     Can the findings of the present study on laboratory animals (rats groomed in controlled setting) be correlated with humans or will there be other factors (environmental, social, etc.) that might play role in humans?

Round 2

Reviewer 1 Report

14 February 2023 

Manuscript ID: brainsci-2175781

Type: Article

Title: ‘Effects of electroacupuncture on prefrontal cortex and liver of rats exposed to CUMS through P2X7R/NLRP3/IL-1β’ by Pang et al., submitted to Brain Sciences 

Dear Authors,

I am pleased to see that the authors took my comments seriously and have solved most of the issue I raised in the previous round. Currently, the manuscript is a well written and nicely presented research article studying the effects of electroacupuncture on the prefrontal cortex and liver of rats subjected to chronic unpredictable mild stress. That said, I just leave some comments here, which, I believe, help the authors refine the manuscript to close my part of the peer review session.

1.      Title: Please present the title self-explanatory and stating the most important message of this study and avoid using abbreviation if possible. “The effects” makes the title obscure. Please include the concrete and the most important findings of this study.

2.      Keywords: Please list ten keywords and use as many as possible in the title and in the first two sentences of the abstract

3.      Introduction: This section has improved substantially. Nevertheless, describing the following topics as parts of the main constructs of this paper certainly enriches the introductory section, such as mitochondrial impairment as a common motif in neuropsychiatric presentation, influencers in action in inflammation and kynurenines, and integrating armchair, bench, and bedside research for neuropsychiatry.

4.      Results: I recommend presenting figures in color.

5.      References: Please list references according to the Journal’s guidelines (https://www.mdpi.com/journal/brainsci/instructions): add a period (. ) after abbreviated journal names and place ido numbers.

Overall, the manuscript contains seven tables, four figures and 78 references. I believe that the manuscript carries important value in describing how electroacupuncture may alleviate depressive like behavior and changes in the liver and prefrontal cortex of rats exposed to chronic unpredictable mild stress by inhibiting P2X7R/NLRP3/IL-1β expression.

Best regards,

Reviewer
